

# Environmental inactivation and irrigation-mediated regrowth of *Escherichia coli* O157:H7 on romaine lettuce when inoculated in a fecal slurry matrix

Jennifer A. Chase[1], Melissa L. Partyka[2], Ronald F. Bond[1] and Edward R. Atwill[1]

[1] Western Center for Food Safety, University of California, Davis, Davis, CA, USA
[2] School of Fisheries, Aquaculture and Aquatic Sciences, Auburn University, Auburn, AL, USA

## ABSTRACT

Field trials were conducted in July–August and October 2012 to quantify the inactivation rate of *Escherichia coli* O157:H7 when mixed with fecal slurry and applied to romaine lettuce leaves. Lettuce was grown under commercial conditions in Salinas Valley, California. One-half milliliter of rabbit, chicken, or pig fecal slurry, containing an average of $4.05 \times 10^7$ CFU *E. coli* O157:H7 ($C_0$), was inoculated onto the upper (adaxial) surface of a lower leaf on 288 heads of lettuce per trial immediately following a 2.5 h irrigation event. To estimate the bacterial inactivation rate as a function of time, fecal matrix, irrigation and seasonal climate effects, sets of lettuce heads ($n = 28$) were sampled each day over 10 days and the concentration of *E. coli* O157:H7 ($C_t$) determined. *E. coli* O157:H7 was detected on 100% of heads during the 10-day duration, with concentrations ranging from $\leq 340$ MPN/head (~5-log reduction) to $>3.45 \times 10^{12}$ MPN/head (~5-log growth). Relative to $C_0$, on day 10 ($C_{t=12}$) we observed an overall 2.6-log and 3.2-log mean reduction of *E. coli* O157:H7 in July and October, respectively. However, we observed relative maximum concentrations due to bacterial growth on day 6 (maximum $C_{t=8}$) apparently stimulated by foliar irrigation on day 5. From this maximum there was a mean 5.3-log and 5.1-log reduction by day 10 ($C_{t=12}$) for the July and October trials, respectively. This study provides insight into the inactivation and growth kinetics of *E. coli* O157:H7 on romaine lettuce leaves under natural field conditions. This study provides evidence that harvesting within 24 h post irrigation has the potential to increase the concentration of *E. coli* O157:H7 contamination, if present on heads of romaine lettuce; foliar irrigation can temporarily stimulate substantial regrowth of *E. coli* O157:H7.

## INTRODUCTION

Consumption of contaminated fresh produce continues to be a significant cause of foodborne illness in the US (*Richardson et al., 2017*). For example, in 2015 there were

Corresponding author
Edward R. Atwill,
ratwill@ucdavis.edu

383 reported foodborne illnesses associated with leafy green vegetables, according to the US Centers for Disease Control and Prevention (*Centers for Disease Control and Prevention (CDC), 2017*). Despite a considerable amount of effort by private industry, academia, and governmental agencies to fully understand the core mechanisms and contributing factors responsible for microbial contamination in the preharvest environment, it is not known what proportion of the annual foodborne incidence of *Escherichia coli* O157:H7 or other pathogenic enteric bacteria are associated with produce is due to contaminated irrigation water, wildlife intrusion, or the use manure-based soil amendments in the preharvest production environment.

Research studies have been conducted to estimate the prevalence of pathogens in irrigation water supplies (*Falardeau et al., 2017*; *Gorski et al., 2014*; *Partyka et al., 2018b*; *Ssemanda et al., 2018*), survival kinetics of these pathogens in agricultural water (*Steele & Odumeru, 2004*; *Wang et al., 2018*), and how well pathogens survive once transferred onto crops via irrigation (*Atwill et al., 2015*; *Chase et al., 2017*; *Moyne et al., 2011*; *Van Der Linden et al., 2014*; *Weller et al., 2017b*). For example, numerous studies have shown the occurrence of bacterial pathogens in US agricultural water supplies, especially for surface water sources (*Falardeau et al., 2017*; *Partyka et al., 2016*, *2018a*, *2018b*). While contaminated irrigation water can act as a vehicle to directly transfer pathogens onto crops (*Barker-Reid et al., 2009*; *Erickson et al., 2010*; *Fonseca et al., 2011*; *Solomon, Pang & Matthews, 2003*; *Solomon, Potenski & Matthews, 2002a*; *Wachtel, Whitehand & Mandrell, 2002*), it has also been shown to transfer pathogens from either manure or manure-amended soil (*Franz, Semenov & Van Bruggen, 2008*; *Mootian, Wu & Matthews, 2009*; *Solomon, Yaron & Matthews, 2002b*); and simulated wildlife scat onto nearby lettuce heads (*Atwill et al., 2015*; *Weller et al., 2017a*).

These studies indicate that pathogen transfer onto crops can occur under specific conditions; less clear is the survival kinetics of these transferred pathogens once on the surface of produce and whether a sufficient proportion of pathogens survive to the point of harvest and thereby manifest as a food safety hazard. Pathogens such as *E. coli* O157:H7 deposited onto produce via irrigation water have been shown to experience high levels of log-reduction under summer field conditions (*McKellar et al., 2014*; *Moyne et al., 2011*). However, a recent study under California summer conditions found that *E. coli* O157:H7 mixed with a fecal slurry to simulate fecal splash during foliar irrigation exhibited reduced rates of inactivation in contrast to this earlier work (*Chase et al., 2017*). In addition, *Weller et al. (2017b)* replicated these findings of extended survival of *E. coli* when the bacteria were mixed in a fecal slurry under growing conditions typical of upstate New York.

The present follow-up study was conducted to further characterize survival kinetics of *E. coli* O157:H7 in wildlife fecal slurries applied to romaine lettuce to simulate fecal splash caused by foliar irrigation, stratified by vertebrate sources of feces (rabbit, pig, chicken) and growing seasons (July–August and October). In addition, the effect of irrigation on the population of inoculated bacteria was investigated by irrigating midway through the trial (day 5) to determine if irrigation had a significant effect on pathogen persistence on romaine lettuce leaves.

## MATERIALS AND METHODS

### Field site and lettuce plot

During the summer (July–August) and fall (October) of 2012, romaine lettuce (*Lactuca sativa* L. var. *longifolia*) was grown under commercial conditions in the Salinas Valley, California with plant thinning, weed removal and irrigation being performed as needed up to field inoculation described by *Chase et al. (2017)*. The lettuce was planted on June 12 and August 14 for the summer and fall trials, respectively. At the time of inoculation, lettuce plants were 60 days old from in-ground seed sowing and were in the head formation stage. The field plot (18.3 m wide × 90.5 m long) consisted of beds 56–61 cm wide separated by furrows 40–46 cm wide; each bed contained two parallel rows of lettuce separated by 30.5 cm at time of planting. A subplot of 20 adjacent beds within the experimental field plot were allocated for this bacterial inactivation study. Lettuce heads were randomly selected for inoculation and identified by placing a color-coded stake flag, labeled with the lettuce head collection ID, directly behind the head. Prior to slurry inoculation on July 27, 2012 and October 16, 2012, the field was irrigated for roughly 3.5 h and 2.5 h with a mean cumulative water application of 3.35 cm and 2.41 cm, respectively. Field maintenance (e.g., irrigation, plant thinning, weed removal) were halted for the remainder of the experiment, apart from a planned mid-trial foliar irrigation event. The mid-trial foliar irrigation event occurred in the morning, 5 days post-inoculation for $2.65 \pm 0.12$ h during which an average 2.59 cm and 2.41 cm of water was applied on August 1, 2012 and October 21, 2012, respectively. The amount of water applied during each irrigation event was determined using an online irrigation and management tool Crop Manage available at https://cropmanage.ucanr.edu.

### Weather data

Hourly meteorological data was collected through the California Irrigation Management Information System (CIMIS), including; cumulative solar radiation ($W/m^2$), air temperature (°C), evapotranspiration (ETo, mm), relative humidity (%), wind speed (m/s) and direction (°), soil temperature (°C) and rainfall (mm) (Table S1). CIMIS station 116 (latitude 36°43′1.0″N, longitude: −121°41′31.0″W) was chosen based on proximity and by the recommendation from the ETo Zone brochure available on the CIMIS website (www.cimis.water.ca.gov), which was ~2.5 km from the study plots (latitude: 36°37′41.8″ N, longitude 121°32′26.4″W). An additional HOBO weather station was placed within the lettuce plots and leaf wetness (%) was recorded in 15-min intervals. No recorded precipitation occurred during either trial. However, typical to the area, there was an early morning marine layer that gradually cleared by mid-day due to seasonal prevailing northwest wind patterns in the Salinas Valley; the winds increase during the day due to the pinching of airflow through the Mount Diablo and Santa Lucia mountain ranges.

### Bacterial strain

We used a rifampicin resistant (50 μg/mL) *E. coli* O157:H7 strain ATCC 700728 that lacked *Stx* one and two genes (*Atwill et al., 2015*; *Chase et al., 2017*; *Moyne et al., 2011*). Bacterial cultures were grown in tryptic soy broth (TSB) (Difco, BD, Sparks, MD, USA)

supplemented with 50 μg/mL rifampicin (+R) (Gold Biotechnology, St. Louis, MO, USA) and incubated at 37 °C for 6 h with orbital rotation (100 rpm). Cell counts were first estimated using a regression equation that extrapolated bacterial concentration from the optical density (OD, 600 nm), then confirmed by serial dilution in phosphate buffer saline (PBS) (Sigma-Aldrich St. Louis, MO, USA), spread plate onto Tryptic Soy Agar (Difco, BD, Sparks, MD, USA) supplemented with +R (TSA+R), and incubated at 37 °C for 18–24 h.

## Fecal slurry and spiking with *E. coli* O157:H7

Fecal material was collected within 24 h of slurry construction from three species: rabbits (*Oryctolagus cuniculus*; New Zealand Whites), chickens (*Gallus gallus domesticus*) and pigs (*Sus scrofa domesticus*) housed at the Teaching and Research Animal Care Services (TRACS) Facility, Hopkins Avian Facility, and the Swine Teaching and Research Center, respectively, all located at the University of California, Davis. These three sources of fecal material were selected for inclusion in this study because growers and researchers have consistently observed these wildlife species (avian, rodents, feral pigs) intruding into and defecating on agricultural fields of central coastal California. In addition, using laboratory animal populations housed at UC Davis allowed us to reasonably control for fecal consistency and mammalian source of feces. The rabbit cages and chicken coop had been cleaned within 12 h of collection. The pig fecal material was freshly voided within 2 h of collection. Fecal samples were pooled from multiple individuals housed in their respective TRACS facilities. Feces used in the experiments were confirmed negative for *E. coli* O157:H7 and rifampicin resistant non-O157 *E. coli*. Fecal slurries (rabbit, chicken, or pig) were constructed as described in *Chase et al. (2017)*, modified using a fecal-PBS ratio of 1 g:4 mL. Briefly, feces and PBS were stomached in a 55-oz Whirl-Pak filter bag (Nasco, Fort Atkinson, WI, USA) using a Seward Stomacher 80 Lab Blender (Seward Laboratory Systems Inc., Bohemia, NY, USA) for 5 min at 230 rpm; the liquid portion was dispensed into a sterile flask, vortexed for 1 min, then inoculated with the target concentration of *E. coli* O157:H7. Prior to use, the slurries were homogenized by vigorous shaking for 1 min. Slurries were stored at 4 °C for approximately 12 h prior to field inoculations. *E. coli* O157:H7 concentrations were confirmed using the serial dilution spread plate method described above in triplicate per slurry type; the concentration in each slurry was estimated immediately following construction and again immediately before field inoculations to confirm no significant growth or death occurred during the 12-h refrigeration hold.

During each trial, 288 lettuce heads were inoculated with one of the three fecal slurries (96 heads per fecal type); the slurry was applied directly onto the adaxial (upper) surface of a lower outer leaf. During the July and October trials, the 0.5 mL of slurry inoculum contained an average of $3.90 \times 10^7$ and $4.20 \times 10^7$ CFU *E. coli* O157:H7, respectively (Table 1).

## Collection of lettuce

A total of 12 collection events occurred over a 10-day (236 h) period. Collection times varied slightly over the two fields trials; the average collection time (h) since inoculation

**Table 1 Mean concentrations and log-reductions of *E. coli* O157:H7[a] recovered on romaine lettuce over two 236 h field trials in central coastal California, 2012.**

| Fecal type | % pos.(*n*) | Applied CFU/head ($C_0$) | $C_{t=1}$ day 0.1 MPN/head | $C_{t=6}$ day 5 MPN/head | $C_{t=8}$ day 6 MPN/head | $C_{t=12}$ day 10 MPN/head | log($C_{t=12}/C_0$) | log($C_{t=12}/C_{max}$)[d] |
|---|---|---|---|---|---|---|---|---|
| July 27–Aug 6, Irrigation on Aug 1, 1.3–3.9 cm[c] | | | | | | | | |
| Pig | 100% (95[b]) | $3.78 \times 10^7$ | $7.64 \times 10^8$ | $3.31 \times 10^8$ | $8.76 \times 10^{11}$ | $6.29 \times 10^4$ | −2.78 | −7.14 |
| Rabbit | 100% (96) | $3.76 \times 10^7$ | $7.11 \times 10^8$ | $7.99 \times 10^6$ | $4.10 \times 10^{10}$ | $7.51 \times 10^4$ | −2.70 | −5.74 |
| Chicken | 100% (96[b]) | $4.17 \times 10^7$ | $1.59 \times 10^8$ | $7.69 \times 10^6$ | $4.31 \times 10^{11}$ | $1.53 \times 10^5$ | −2.44 | −6.45 |
| Oct 16–Oct 26, Irrigation on Oct 21, 1.3–2.2 cm[c] | | | | | | | | |
| Pig | 100% (94[b]) | $4.56 \times 10^7$ | $1.02 \times 10^8$ | $1.53 \times 10^5$ | $8.75 \times 10^9$ | $1.24 \times 10^4$ | −3.56 | −5.85 |
| Rabbit | 100% (96[b]) | $5.00 \times 10^7$ | $6.72 \times 10^8$ | $2.99 \times 10^6$ | $4.35 \times 10^7$ | $4.27 \times 10^4$ | −3.07 | −3.01 |
| Chicken | 100% (95[b]) | $3.03 \times 10^7$ | $4.95 \times 10^8$ | $5.77 \times 10^5$ | $1.01 \times 10^8$ | $1.82 \times 10^4$ | −3.22 | −3.74 |

Notes:
[a] Data for day 0.1, 5, 6, and 10 post-inoculation shown.
[b] Samples below the limit of detection were assigned a value of 248 MPN/head, samples exceeding the limit were assigned a value of $3.45 \times 10^{12}$ MPN/head.
[c] Cumulative volume (cm) range of applied irrigation water during the mid-trial irrigation events occurring on day 5, measured with rain gauges placed throughout the field.
[d] $C_{max}$ occurred on day 6 of all trials ($t = 8$).

($t$ = event:day) included: $0.5 \pm 0.3$ h ($t = 1{:}0.02$), $21.8 \pm 0.8$ h ($t = 2{:}0.9$), $44.6 \pm 0.1$ h ($t = 3{:}1.9$), $68.9 \pm 0.4$ h ($t = 4{:}2.9$), $93.3 \pm 0.5$ h ($t = 5{:}3.9$), $115.9 \pm 0.1$ h ($t = 6{:}4.8$), $121.3 \pm 1.8$ h ($t = 7{:}5.1$), $141.8 \pm 0.5$ h ($t = 8{:}5.9$), $164.9 \pm 0.4$ ($t = 9{:}6.9$), $188.5 \pm 0.5$ h ($t = 10{:}7.9$), $212.8 \pm 0.3$ h ($t = 11{:}8.9$) and $235.4 \pm 0.6$ ($t = 12{:}9.8$). A total of 24 heads of lettuce were harvested per collection event, with eight heads per fecal slurry type, using methods described by *Atwill et al. (2015)* and *Chase et al. (2017)*. Briefly, lettuce was aseptically harvested and placed in a 24 × 24 Bitran bag (Uline, Pleasant Prairie, WI, USA); after each head was harvested, the sampler put on a new pair of gloves and cleaned the harvesting knife with 70% ethanol. We adhered to a strict QAQC protocol using both laboratory and field controls; in addition to using laboratory positive and negative controls, negative field control heads (*n* = 4) were also collected each day from a negative control plot immediately adjacent to the experimental plot and processed using the same protocol as the inoculated heads. To confirm that the irrigation water was not a source of the target organism, 20 L of water was collected from a single sprinkler head in sterile carboy (Nalgene Nunc, Penfield, NY, USA) during each irrigation event (*n* = 4). Inoculated and negative control lettuce samples, as well as the water samples, were placed on ice (~4 °C) and transported to UC Davis for analysis.

## Bacterial enumeration and confirmation of *E. coli* O157:H7

Samples were processed upon arrival at the laboratory, within 4 h ± 0.5 h of collection. Samples were processes using methods previously described by *Atwill et al. (2015)* and *Chase et al. (2017)* which is designed to enumerate a wide range of bacterial concentrations; with the lower limit of enumeration being 0.68 MPN/mL washate (equivalent to 340 MPN/whole head of lettuce) and the upper limit of enumeration being $6.9 \times 10^9$ MPN/mL washate (equivalent to $3.45 \times 10^{12}$ MPN/whole head of lettuce). Briefly, a 500 mL PBS+R washate was added to each sampling bag containing a head of lettuce. Samples were lightly massaged and vigorously shaken to detach *E. coli* O157:H7

cells from the lettuce surface into solution. The concentration of *E. coli* O157:H7 was determined by transferring one mL of the PBS washate into the first two positions of a 12-well-deep reservoir (VWR, Radnor, PA, USA) containing nine mL TSB+R, followed by duplicate 100-fold serial dilutions in the remaining 10 wells containing 9.9 mL TSB+R. Sample reservoirs were incubated at 37 °C for 24 h with orbital rotation of 50 rpm. The sample enrichments were channel struck, using a 12-channel pipette, onto CHROMagar O157 (CHROMagar, Paris, France) supplemented with +R and incubated at 37 °C for 24 h. Bacterial concentrations were estimated using two MPN calculators (*Curiale, 2009*; *Jarvis, Wilrich & Wilrich, 2010*) then extrapolated from a per-mL to a per-head outcome by multiplying by 500 (volume of washate per head of lettuce). The sensitivity of this MPN assay was determined using laboratory positive controls, set up as triplicate 10-fold serial dilutions ranging from $5.15 \times 10^6$ CFU to $5.15 \times 10^1$ CFU of *E. coli* O157:H7. Each bacterial suspension was inoculated onto a head of lettuce, held for 4 h under refrigeration to account for transportation time, and the assay performed as described above. All positive control heads of lettuce tested positive, and the MPN estimates generally matched the expected values of the 10-fold serial dilutions ($5.15 \times 10^6$ CFU to $5.15 \times 10^1$ CFU of *E. coli* O157:H7).

Isolates from a subset of samples ($n = 60$) were confirmed by PCR, followed by pulsed-field gel electrophoresis to confirm isolates matched our original inoculum, as described by *Atwill et al. (2015)* and *Chase et al. (2017)*. Water samples were analyzed using ultrafiltration methods described in *Partyka et al. (2018a)* and all samples were confirmed negative.

## Statistical analysis

Negative binomial regression was used to model the daily concentration of *E. coli* O157:H7 on heads of romaine lettuce over the two 10-day trials using STATA® 14 software (StataCorp LP, College Station, TX, USA). The outcome of interest was the estimated (MPN) concentration of *E. coli* O157:H7/head ($C_t$) as a function collection event ($t$) with the inclusion of an offset variable ($C_0$) to account for the varying inoculum concentrations described in Table 1. Outcomes that exceeded the bounds of detection using the MPN methods described above were excluded from analysis: 10 samples were positive only after enrichment (<340 MPN/head) and three samples were TNTC ($>3.45 \times 10^{12}$ MPN/head). Additional covariates and interaction terms were included to help explain the variability in our observed data. In this study we adhered to a detailed model-fitting protocol as described by *Partyka et al. (2017)*: final models were selected based on a combination of link-test, likely-hood ratio test, AIC score and pseudo-$R^2$. The link-test was used to confirm that our dependent variable was properly specified, which resulted in a significant linear predictor ($P = 0.000$) and insignificant ($P = 0.348$) linear predictor squared.

## RESULTS

A total of five hundred and seventy two ($n = 572$) inoculated heads of lettuce were collected and processed during two separate 10-day (236 h) field trials in July–August ($n = 287$)

**Table 2 Negative binomial regression model[a] for the rate of inactivation of E. coli O157:H7[b] on heads of romaine lettuce (n = 559) over 10 days in central coastal California during July–Aug. and Oct., 2012.**

| Factor | Factor Categories | Coefficient | Std. Err. | 95% C.I. | P-value |
|---|---|---|---|---|---|
| Fecal | Pig[c] | 0.0 | | | |
| | Rabbit | −1.232 | 0.334 | −1.888, −0.576 | 0.000 |
| | Chicken | −1.289 | 0.342 | −1.959, −0.619 | 0.000 |
| Trial | July[c] | 0.0 | | | |
| | October | −4.066 | 0.375 | −4.801, −3.331 | 0.000 |
| Trial × Fecal | July × Pig[c] | 0.0 | | | |
| | October × Rabbit | 1.929 | 0.487 | 0.974, 2.884 | 0.000 |
| | October × Chicken | 2.016 | 0.480 | 1.076, 2.956 | 0.000 |
| Daily leaf wetness[d] | | 0.041 | 0.012 | 0.017, 0.065 | 0.001 |
| Sampling event[e] | 1 (0.5 ± 0.3)[c] | 0.0 | | | |
| | 2 (21.8 ± 0.8) | −0.657 | 0.391 | −1.423, 0.109 | 0.093 |
| | 3 (44.6 ± 0.1) | −1.565 | 0.490 | −2.526, −0.605 | 0.001 |
| | 4 (68.9 ± 0.4) | −4.170 | 0.408 | −4.97, −3.371 | 0.000 |
| | 5 (93.2 ± 0.5) | −5.642 | 0.408 | −6.441, −4.842 | 0.000 |
| | 6 (115.9 ± 0.1) | −3.880 | 0.450 | −4.762, −2.998 | 0.000 |
| | 7 (119.5 Oct.)[f] | −6.167 | 0.517 | −7.179, −5.154 | 0.000 |
| | 7 (123.0 July)[f] | −0.133 | 0.655 | −1.417, 1.150 | 0.839 |
| | 8 (141.7 ± 0.5) | 3.215 | 0.462 | 2.308, 4.121 | 0.000 |
| | 9 (164.9 ± 0.4) | −5.130 | 0.426 | −5.965, −4.295 | 0.000 |
| | 10 (188.5 ± 0.5) | −7.180 | 0.419 | −8.002, −6.359 | 0.000 |
| | 11 (212.8 ± 0.3) | −6.274 | 0.462 | −7.179, −5.369 | 0.000 |
| | 12 (235.4 ± 0.6) | −9.230 | 0.396 | −10.005, −8.454 | 0.000 |
| Intercept | | 2.997 | 0.754 | 1.520, 4.474 | 0.000 |

Notes:
[a] Regressed onto $C_t$ as the outcome variable.
[b] Bacteria concentrations were estimated using MPN methods.
[c] Referent.
[d] Based on the averaged HOBO data collected in 15-min. intervals beginning 24 h prior to harvest.
[e] Categorized time since inoculation with actual hours displayed in parenthesis.
[f] Event 7 was modeled as two separate events to capture the 4.5 h collection lag between both trials.

and October ($n = 285$) of 2012; four samples were excluded from processing due to faulty collection bags. Concentrations of *E. coli* O157:H7 in the 0.5 mL inocula per head of lettuce ($C_0$) were similar between both trials, ranging from $3.76 \times 10^7$ to $4.17 \times 10^7$ CFU for the July trial and $3.03 \times 10^7$ to $5.00 \times 10^7$ CFU in the October trial. Results of laboratory analyses were recorded as MPN/head of lettuce ($C_t$) and standardized for statistical analysis using the offset option for negative binomial regression to account for variability in initial inoculum concentration ($C_0$) shown in Table 2. Nearly all heads of lettuce (559/572, 97.7%) had quantifiable concentrations of bacteria during the 10-day trial; in 10 samples, *E. coli* O157:H7 were detectable only after enrichment (<340 MPN/head, lower enumeration limit) and the upper enumeration limit (>$3.45 \times 10^{12}$ MPN/head) was exceeded in three samples (Table 1). This indicates that none of the heads of lettuce experienced greater than 7.5- to 7.7-log reduction relative to $C_0$ during the pair of 10-day
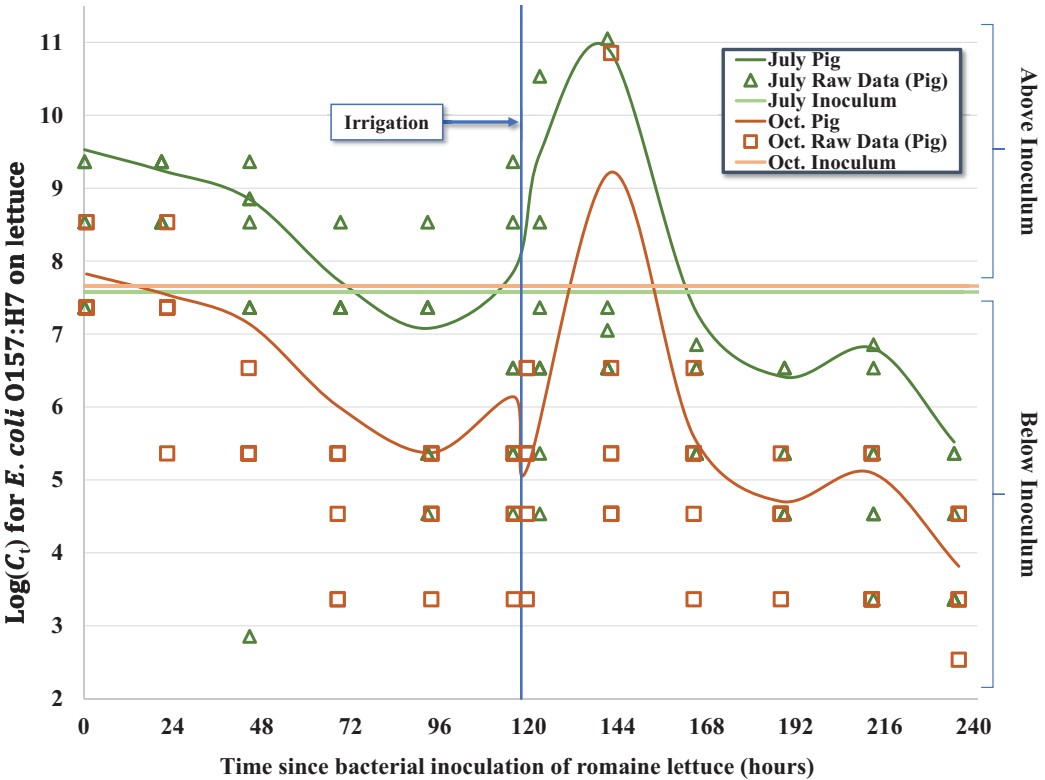

**Figure 1 Regression model for the inactivation rate of *E. coli* O157:H7 in pig fecal slurry on romaine lettuce grown in central coastal California during two 10-day field trials occurring in 2012, holding leaf wetness constant using the median values of 33% and 32% for the July–Aug. and Oct. trials, respectively.**

trials for three fecal matrices (pig, rabbit, chicken) and two seasons, with many heads of lettuce experiencing substantially less reductions in *E. coli* O157:H7. In terms of general trends averaged across the three fecal matrices, we observed a mean 2.6- and 3.2-log reduction compared to initial concentrations ($C_0$) in July and October, respectively, but given the regrowth of *E. coli* O157:H7 following foliar irrigation discussed below, the mean log reduction on day 10 relative to $C_{max}$ on day 6 ($C_{t=8}$) ranged from 5.1 to 6.7 (Table 1; Fig. 1).

Hourly weather data were retrieved from two nearby CIMIS weather stations beginning 24 h before the start of each trial and continuing through the final day of collection to calculate daily averages and ambient conditions during the 24 h before collection (Table S1). Prior to completion of the October 2012 trial, the closest CIMIS weather station (#89) was deactivated, requiring use of an alternative station (CIMIS #116) more than two km away. For consistency, only data from site #116 were utilized in comparisons between the trials for weather characteristics, in addition leaf wetness was collected by the onsite HOBO station. There was no measurable precipitation recorded during either trial. Average air temperature (°C) during the trials was significantly higher in October ($P < 0.0002$) than July, with higher maximum ($P < 0.0001$) and lower minimum temperatures ($P < 0.0001$) (Table S1). Although long summer days in July produced

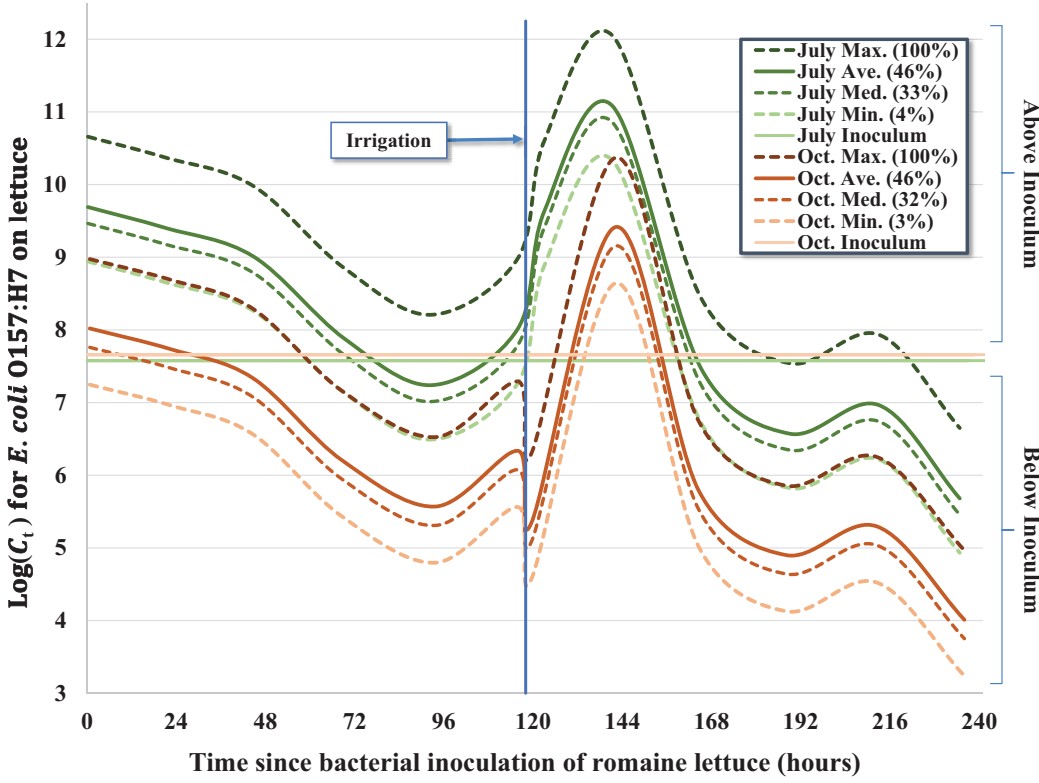

**Figure 2 Predicted effect of leaf wetness on the inactivation rate of *E. coli* O157:H7 in pig fecal slurry on heads of romaine lettuce in central coastal California.**

higher average solar radiation (W/m$^2$) ($P < 0.0046$) than recorded in October, July also experienced multiple days of dense morning fog leading to significantly higher average humidity ($P < 0.0001$) and warmer soil temperatures ($P < 0.0001$) and overall cooler weather (Table S1). Air temperature, averaged across days-since-inoculation, was positively associated ($P < 0.0001$) with log-transformed *E. coli* O157:H7 concentrations for both July ($\beta = 0.54$, $R^2 = 0.08$) and October ($\beta = 0.44$, $R^2 = 0.33$); however, all weather variables except for leaf wetness were non-significant ($P > 0.05$) in the final model of *E. coli* O157:H7 concentration on heads of romaine lettuce, when collection event was introduced as a covariate (Table 2). This lack of significance for most weather variables when time-since-inoculation was included in the regression model is likely due to collinearity between time-since-inoculation and these other weather variables.

Despite similar starting concentrations ($C_0$) for the July–August and October trials (Table 1), there was a significant reduction in the concentration of *E. coli* O157:H7 throughout the duration of the October trial relative to the earlier trial, which we hypothesize is the consequence of bacterial growth during cool, wet conditions during the early phase of the July–August trial. The average 24-h leaf wetness was positively associated with *E. coli* O157:H7 concentration throughout the trial; each additional percent increase of wetness was associated with ~4.2% ($e^{0.041 \times 1} = 1.042$) increase in the concentration of *E. coli* O157:H7 per lettuce head (Fig. 2). We observed a substantial increase of *E. coli* O157:H7 during the initial 2 days ($\leq 44.75$ h) post-inoculation for both trials, but especially during the

July–August trial. Approximately 55% ($n = 37$) of the inoculated heads of lettuce exhibited higher bacterial concentrations consistent with bacterial growth relative to the original inoculum dosage ($C_0$) during the first 2 days of the July trial, while only ~24% ($n = 17$) of heads of lettuce exhibited bacterial concentrations above $C_0$ in the October trial. This difference in bacterial growth in the early part of the July trial compared to October resulted in a significant seasonal effect in the regression model (Table 2), with $C_{t>0}$ values in October having on average ~98.3% less *E. coli* O157:H7 ($e^{-4.066 \times 1} = 0.017$) compared to the July–August trial. During the cooler and wetter July trial, lettuce heads inoculated with the pig slurry accounted for the majority (42%, $n = 25$) of total lettuce samples above $C_0$, with 32% ($n = 19$) and 26% ($n = 15$) of lettuce heads above $C_0$ inoculated with rabbit and chicken slurry, respectively. However, during the October trial, lettuce heads inoculated with the pig slurry accounted for a minority 19% ($n = 4$) of the samples with bacterial concentrations above $C_0$, while 33% ($n = 7$) and 48% ($n = 10$) of lettuce heads exhibiting bacterial concentrations above $C_0$ were inoculated with rabbit and chicken slurry, respectively. This inverse relationship between fecal matrix source and the proportion of lettuce heads exhibiting growth above $C_0$, explains the significant interaction term of fecal matrix $\times$ trial included in the final model (Table 2): both rabbit and chicken fecal matrix terms have >2.0 coefficients for their interaction with October, resulting in a combined coefficient (main effect + interaction term) greater than zero, which is the referent value for pig slurry. Differences in the microbiota and nutritional composition of the mammalian fecal material used between the two trials (i.e., different laboratory animals) may help explain the differing effects of fecal matrix between trials.

Given the complex nature of how *E. coli* O157:H7 behaved on the surface of lettuce leaves as a function of time-since-inoculation, fecal matrix and environmental or weather variables, multiple regression models were considered. For example, regression equations with quadratic and higher order variables for time-since-inoculation as a continuous variable were considered to adequately describe bacterial populations ($C_t$) on each head of lettuce over time. However, setting time-since-inoculation as a categorical variable allowed a better fit to the complex raw data, especially given the significant multi-log increase in $C_t$ values immediately after the irrigation event and reaching a maximum observed concentration on day 6 (Tables 2 and 3; Fig. 1). For example, the average $C_{t=6}$ immediately before irrigation (day 5) and 1-day post-irrigation ($C_{t=8}$, day 6) went from $1.11 \times 10^8$ to $4.50 \times 10^{11}$ MPN/head and from $1.24 \times 10^6$ to $3.23 \times 10^9$ MPN/head in July and October, respectively (Table 1). Irrigation-mediated bacterial regrowth was not consistent across slurry matrices or trials; for pig, rabbit and chicken we observed a mean regrowth ($\log_{10} (C_{t=6 \text{ (pre-irrigation)}}/C_{t=8})$) of 3.4-, 3.7- and 4.8-log for the July samples compared to 4.8-, 1.2- and 2.3-log in October, respectively. We speculate that there were shifts in the slurry microbiota or interactions between the slurry type and microclimates that could have contributed to these fluctuations. The peak in $C_t$ values on day 6 ($C_{t=8}$) was quickly followed by a multi-log reduction on day 7 ($C_{t=9}$) to concentrations similar to days 4 and 5 ($C_{t=4}$ and $C_{t=5}$) prior to foliar irrigation, followed by bacterial tailing and brief regrowth (day 9, $C_{t=11}$) during the remainder of the 10-day trial (Table 1; Fig. 1). The subtle yet significant

phase of <1-log regrowth 4 days after foliar irrigation (day 9, $C_{t=11}$) was observed in both the July–August and October trials.

## DISCUSSION

This project was the continuation of a multistate and multiyear effort coordinated by FDA to examine how local environmental factors across the US may influence the survival dynamics of fecal bacteria on lettuce under field conditions in order to better understand and predict potential food safety risks from consumption of produce. A study with similar design was conducted in upstate New York by *Weller et al. (2017b)* using methods based on *Atwill et al. (2015)*, *Chase et al. (2017)*, and this trial, in an effort to examine the behavior of bacterial pathogens under different field conditions. However, in the *Weller et al. (2017b)* study, a cocktail of rifampicin-resistant *E. coli* was used instead of the *E. coli* O157:H7 strain used in this study. In contrast to our current study which observed an overall average inactivation rate of 0.28 log MPN day$^{-1}$, *Weller et al. (2017b)* observed a higher inactivation rate of 0.52 log MPN day$^{-1}$ that was similar to the ~0.6 log MPN day$^{-1}$ observed by *Chase et al. (2017)*. However, when the maximum recovered concentrations occurring at day 6 post-irrigation ($C_{t=8}$) was used as the baseline for calculating the inactivation rate, the observed inactivation rate for this study was ~1.32 log MPN day$^{-1}$ or three-fold higher compared to *Weller et al. (2017b)*. These differences in the inactivation rates may be due to one or more different factors between the New York and the current California trial: bacterial isolates and starting concentrations, soil composition, field design, irrigation practices and weather conditions. For instance, foliar irrigation was used in our trial while in contrast, a rainfall event occurred in the New York trial, with substantial differences in the rate and total volume of received water. Our California trial applied approximately 3.7× and 3.4× more water in July–August and October, respectively, compared to the largest rain event in the *Weller et al. (2017b)* trial. Water droplet size and application rates could also affect bacterial detachment and/or growth trajectories; the rain event described by *Weller et al. (2017b)* occurred over 5 h with an intensity of 1.42 mm rain/h while the California irrigation events occurred over 2.5 h with a 9.35 and 9.65 mm/h application rate in July–August and October. *Weller et al. (2017b)* found that the occurrence of a precipitation event increased the rate of inactivation from 0.32- to 1.08-log *E. coli* MPN day$^{-1}$; we observed a rapid increase in bacterial concentration immediately following inoculation followed by a gradual decline of 0.11-log *E. coli* O157:H7 MPN day$^{-1}$ mean inactivation prior to irrigation ($t$ = 1 through $t$ = 5), with an average inactivation rate of 0.76-log *E. coli* O157:H7 MPN day$^{-1}$ following irrigation ($t$ = 6 through $t$ = 12). This substantial difference in the inactivation rate of *E. coli* on heads of lettuce between California and New York trials underscores the importance for collating or comparing data from different growing regions across the US in order to better model the regional risks of foodborne illness associated with the consumption of raw or minimally processed produce.

Over the 10-day trial, we observed significant departure from a first-order inactivation process across all fecal matrices (pig, rabbit, chicken) and seasons (July–August and October), in part caused by multi-log regrowth of bacteria following foliar irrigation.

As a result, when modeling these kinetics, we used a categorized time variable to fully capture the time-dependent inactivation and regrowth of *E. coli* O157:H7 (Fig. 1). Nonetheless, some general features of bacterial inactivation are evident. Although a relatively uniform slurry inoculum was generated between fecal matrices and season (Table 1), a wide range of bacterial concentrations were recovered during each sampling event with similar variability across both trials (Fig. 1), consistent with the findings reported by *Chase et al. (2017)*. Apparently, despite the lettuce being in the same plot and inoculated with the same strain of *E. coli* O157:H7, either the microenvironment, leaf surface microbiome, availability of nutrients, exposure to solar radiation, or other factor(s) differed enough from head to head that we observed a 4- to 6-log difference in $C_t$ within a few days of inoculation (Fig. 1). This level of variability in the level of *E. coli* O157:H7 on lettuce heads could translate to a wide variability in risk of illness from consumption of these lettuce heads, at least three orders of magnitude based on the dose response function of *Teunis, Ogden & Strachan (2008)*. The actual risk of human infection from consuming these heads of lettuce will also be dependent on factors such as trimming outer leaves during harvest, field packing method, retail and consumer handling practices and serving size that collectively impact the final ingested dose relative to $C_t$.

Negative binomial regression is a relatively contemporary application for modeling this type of food microbiology data, but has been used successfully to model microbial concentrations for a variety of food matrices and environmental samples, such as *E. coli* O157:H7 concentrations on heads of lettuce due to fecal splash from foliar irrigation (*Atwill et al., 2015*) and survival (*Chase et al., 2017*), concentrations of *Giardia duodenalis* in dairy farm runoff (*Miller et al., 2007*), and *E. coli* concentrations in irrigation water from leafy green produce farms (*Benjamin et al., 2013*). Negative binomial regression is well suited to handle the variance of highly right-skewed and overly dispersed bacterial count data (e.g., Fig. 1) compared to Poisson or normal (Gaussian) regression which either constrains the variance to equal the mean (*Hilbe, 2008*) or requires the variance to be independent of the mean and homoscedastic, respectively. Equally important, negative binomial regression avoids the use of log-transformed data compared to log-normal regression common to many analyses in food science (*McKellar et al., 2014*); analyses based on log-transformed data can be negatively biased when used to make microbial predictions on non-log-transformed data due to the phenomena known to some as Jensen's Inequality (*Casella & Berger, 1990*; *Feng et al., 2013*).

*Medicine & Council (2003)* indicate that caution should be used when utilizing statistical models to analyze dynamic bacterial data with the goal of informing food safety policies. *Fakruddin, Mazumder & Mannan (2011)* states that overall trends in microbial population data can be explained using statistical models, however, the results should not be extrapolated beyond the environmental or temporal conditions defined within a study. Rather than use negative binomial regression for modeling bacterial counts as was done with this study, other investigators such as *McKellar et al. (2014)* and *Weller et al. (2017b)* have used Weibull (survival) analysis to model the rate of change of a bacterial population on a head of lettuce. Survival analysis was designed to observe defined events occurring to *an individual* over a defined time interval (*Gail, 2000*; *Stevenson, 2007*).

Due to short generations times, cell division and uncertainty in tracking individual cells across time to allow for proper calculation of time-in-study, estimating the survival function on replicating bacterial populations would be equivalent to predicting survival functions of an individual based on events of their great $n$th grandchild. This approach of using survival analysis, in addition to the use of negative binomial, Poisson and other such regression methods like log-normal, are effectively interval-censored which can result in an underestimation of the true time-to-bacterial decay (*Abernethy, 1996*; *Cain et al., 2011*; *Hougaard, 1999*; *Kleinbaum & Klein, 2011*). While analytical methods have been developed to handle interval-censored survival data (*Huang & Wellner, 1997*), these methods still require that the status of an event is measured at the individual scale level, which is currently not feasible economically or technically for dynamic bacterial populations in field-based trials.

Averaged across the three fecal matrices, we observed a 2.6- and 3.2-log reduction compared to initial concentrations ($C_0$) in July and October, respectively (Table 1; Fig. 1). A previous study conducted in the Salinas Valley with the same strain of attenuated *E. coli* O157:H7 on romaine lettuce inoculated in a rabbit fecal slurry observed a 2.3-log reduction over 92 h (3.8 days), which equates to ~0.6 log MPN day$^{-1}$ (*Chase et al., 2017*) or about twice as large as the rate found in the current study. This ~two-fold difference in the inactivation rate can be explained in part due to the mid-trial irrigation occurring in this 10-day trial but was absent in the previous 4-day trial by *Chase et al. (2017)*. Moreover, the daily varying concentrations of recovered *E. coli* O157:H7 were significantly associated with leaf wetness (24-h-to-harvest average) leading to the higher order regression equations; the recovered *E. coli* O157:H7 concentrations and leaf wetness followed similar fluctuating trajectories (Table 1; Fig. 1). Interestingly, leaf wetness in July 1-day post-irrigation was significantly higher ($P = 0.0114$) than the average wetness observed in October with the corresponding *E. coli* O157:H7 concentration also significantly higher ($P = 0.0309$) in July. Our findings regarding regrowth of *E. coli* O157:H7 following exposure to either higher levels of leaf wetness or foliar irrigation are consistent with the findings reported by *Scherber, Schottel & Aksan (2009)* who observed bacterial cell membrane reconstruction leading to cell proliferation and colony formation within 1-day of adequate rehydration. We speculate that a portion of the bacterial populations were washed off during irrigation, however, given adequate time after rehydration, the populations grew exponentially as evidenced by the $C_t$ immediately following irrigation in October with an average of $5.91 \times 10^5$ MPN/head compared to the samples collected ~3.5 h post-irrigation during the July trial with an average of $1.60 \times 10^9$ MPN/head (Fig. 1). Interestingly, the average bacterial concentrations collected immediately after irrigation (Oct. $C_{t=7}$) could be used to approximate the concentrations recovered 3.5 h post irrigation (July $C_{t=7}$) using the bacterial growth calculation described by *Todar (2008)*.

$$B_{\text{final}} = b_{\text{initial}} \times 2^{n(\text{gentime})} \rightarrow \left(5.91 \times 10^5\right) \times \left(2^{11}\right) = 1.21 \times 10^9 \cong 1.60 \times 10^9$$

The multi-log regrowth and the ability for bacterial cells to recover with adequate rehydration poses challenges for produce safety risk management. However, our data

also demonstrates, that the potential food safety microbial risk may be eliminated or reduced when adequate time for bacterial inactivation following an irrigation event is given.

Curtis (1943) found that guttation on the surface of lettuce is atypical and instead of forming dew-like droplets, the excreted xylem fluids that contain a mixture of organic and inorganic compounds tend to spread and aggregate over the surface of a lettuce leaf. Brandl & Amundson (2008) reported a near 4-log increase of E. coli O157:H7 concentrations when bacteria were grown for 10 h in extruded compounds collected after the guttation process from romaine lettuce. Warm water-saturated soils coupled with high humidity in the absence of sunlight have been shown to promote guttation (Ivanoff, 1963). Mean nighttime humidity significantly decreased by 13.7% ($P < 0.001$) from the July trial ($\hat{x} = 93.3 \pm 2.1\%$, $n = 41$) compared to the October trial ($\hat{x} = 79.6 \pm 6.8\%$, $n = 45$), and mean soil temperature in the 24 h leading up to irrigation was significantly warmer (4.5 °C, $P < 0.0000$) during the July trial ($\hat{x} = 21.2 \pm 0.3$ °C, $n = 24$) compared to the cooler soils of the October trial ($\hat{x} = 16.6 \pm 0.2$ °C, $n = 24$). Soil water saturation was not measured during our trial, however, about 7.5% more irrigation water was applied in July (2.59 cm) compared to October (2.41 cm). We speculate that the conditions for guttation occurred following irrigation, particularly during the July–August trial which would help explain the significant difference in mean growth observed on day 6 ($t = 8$) between the two trials. Furthermore, it is important to note that the application of irrigation is not uniform within an agricultural field (California Department of Water Resources, 1999), which could also help explain, in part, the observed variability on bacterial concentrations per head of lettuce during these trials.

Putting these findings into a pre-harvest food safety perspective, it has been shown that foliar irrigation can act as a vector to transfer bacterial contamination in the form of fecal splash and/or furrow water splash onto the leaves of adjacent lettuce when scat or other fecal matrices are present in the furrow or on the soil surface of beds of lettuce (Atwill et al., 2015; Monaghan & Hutchison, 2012). Results from this study indicate that E. coli O157:H7, when in the presence of a fecal slurry consistent with fecal splash, can possibly grow an additional 1- to 5-log on lettuce over the initial inoculum concentrations and result in substantial microbial risk if harvest occurs within several days after irrigation. During this experiment, there was substantial pathogen-reduction (>2-log) after 10 days when compared to initial bacterial concentrations, or up to 7-log reduction following the peak concentrations occurring the day after foliar irrigation. While our starting bacterial concentrations were consistent with a worst case contamination scenario and likely unrealistic, this high inoculum level is consistent with concentrations seen in super shedder cattle excreting >$10^4$ E. coli O157:H7 CFU/g feces (Munns et al., 2016). Moreover, the use of high inoculum concentrations primarily influences the value of intercept term in a regression model, not the value of the coefficients for such processes as inactivation or bacterial growth secondary to leaf wetness unless such processes are dependent on initial bacterial concentration. Nonetheless, these results should be interpreted with the appropriate level of caution.

## CONCLUSIONS

With respect to appropriate times to harvest lettuce following such events as wildlife intrusion and discovery of excessive amounts of fecal material in a produce field, our results suggest one must take into account whether there is a potential for fecal splash during foliar irrigation, especially if followed by weather conditions conducive to higher leaf wetness (e.g., dense fog) and/or subsequent foliar irrigation that can rewet leaf surfaces and allow multi-log bacterial regrowth to occur. The results from this study shows that adequate wait-times prior to harvest will significantly reduce the concentration of pathogenic bacteria entering into the post-harvest production system and therefore reducing the likelihood of foodborne outbreaks associated with the consumption of raw or minimally processed leafy green produce.

## ACKNOWLEDGEMENTS

We sincerely thank David Oryang, US Food and Drug Administration, for constructive comments on our study design, data interpretation and the draft manuscript, and thank Jane Van Doren, US Food and Drug Administration, for thoughtful edits to the final draft of this manuscript. We would also like to thank Dr. Linda Harris and Anne-Laure Moyne for initial project planning and coordination of field logistics along with data sharing. We would also like to thank the following UC Davis collaborator who provided the fresh fecal samples used in this experiment: Kerry Mello at TRACS (rabbit), Jackie Pisenti at the Avian Facilities (chicken) and Kent Parker at the Swine Center (pig). In addition, laboratory and field assistance was provided by, Kristine Fernandez, Elaine Wang, Panachon Lor, Claudia Bonilla, Jon Molina, Tran Nguyen, Elizabeth Antaki, Chengling Xiao, Anyarat Thiptara, Fhon Saharuetai, Lexi Fisher, Yingjia Benson, and Melinda Fuabel.

### Funding

This project was funded by contract 5U01FD003572-04 from the US Food and Drug Administration for support of the Western Center for Food Safety, UC Davis. The funders provided constructive comments on our study design, data interpretation, and the draft manuscript.

### Grant Disclosures

The following grant information was disclosed by the authors:
US Food and Drug Administration: 5U01FD003572-04.
Western Center for Food Safety, UC Davis.

### Competing Interests

The authors declare that they have no competing interests.

## Author Contributions

- Jennifer A. Chase conceived and designed the experiments, performed the experiments, analyzed the data, prepared figures and/or tables, authored or reviewed drafts of the paper, approved the final draft.
- Melissa L. Partyka conceived and designed the experiments, performed the experiments, analyzed the data, authored or reviewed drafts of the paper.
- Ronald F. Bond conceived and designed the experiments, performed the experiments, approved the final draft.
- Edward R. Atwill conceived and designed the experiments, performed the experiments, contributed reagents/materials/analysis tools, authored or reviewed drafts of the paper, approved the final draft.

## Data Availability

The raw measurements are provided in File S1 (multiple tabs).

## Supplemental Information

Supplemental information for this article can be found online at http://dx.doi.org/10.7717/peerj.6591#supplemental-information.

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
