# Peer review of "Environmental inactivation and irrigation-mediated regrowth of Escherichia coli O157:H7 on romaine lettuce when inoculated in a fecal slurry matrix"

_PeerJ, doi:10.7717/peerj.6591_

## Round 0.1 · original submission · Minor Revisions

Thank you for your patience, getting reviews completed over the holidays is always tricky. Both reviewers were generally positive about the paper and only have minor suggestions for revisions that should be addressed before the paper can be accepted for publication.

·

Basic reporting

The paper is logical in flow and appropriate reference to previous studies thereby providing context for the research.

Experimental design

The methods are appropriate and detail is of nature that will permit replication of the study.

Validity of the findings

The conclusion generated are supported by the data presented. Authors were cautious to not over generalize the significance of the findings. Indeed, based on this studies and those in the literature it appears that significant variability can be associated wit behavior of STEC on crops within a region and between geographic regions. Makes good case for "additional research" must be conducted.

Additional comments

Not very clear why rabbit, poultry (chicken) and pig manures were used. What would be source of these manures...wild (rabbit) and feral (pig) animals? Application of chicken bedding pellets? Were these types of manure used in other studies (besides Chase et. al., 2017)? Authors may need to provide more information rather than just referring to previous publication.

Why were the inoculated slurries held at 4C? Cold shock the cells? Was this also done in other studies? Why was PBS used for dilution of feces? Seems water would be more appropriate.

Would the type of feces influence survival as much as other factors evaluated? I know that only so much can be tested. Not certain whether information exists form previous studies.

Reviewer 2 ·

Basic reporting

This paper is well written and appropriate references have been cited.

Experimental design

Experimental design is pretty straightforward with no major flaws. The authors referred their previous work but they should briefly describe the methodologies.

Validity of the findings

I don't think this paper is novel but it has useful data.

Additional comments

L25: This study was done in 2012 and I am wondering why it took 6 years to write this paper.
L25: E. coli - name in full.
L117: How many times this strain was grown in the lab?
L127: How did you make sure the feces was fresh?
L127: How many animal fecal samples were collected? Just one for each species?
L133: Did the authors homegenize the fecal slurry?
L150: Please describe the methods briefly here.
L161: 340 - lower detection limit? It doesn't seem that good considering the low dose of O157:H7
L165: What is recovery rate of this approach?
L166: Provide data as supplemental.
L169: How many isolates were confirmed? 10% is vague.
Table 1: very basic. should be presented in the supplementary materials.

---

## Round 0.2 · accepted · Accept

Thank you for taking the time to address the reviewers' concerns and clarify the methodological details.

#